# The effect of behavioral activation play therapy in adolescents with depression: A study protocol for a randomized controlled trial

Xiaolong Huang[1,2¤a], Yuqi Chen[3¤b], Jiacheng Luo[4¤c], Dongdong Wang[1,2], Chanjuan Yang[1,2]*, Wei Luo[1,2]*, Yanling Zhou[1,2]*

1 The Affiliated Brain Hospital of Guangzhou Medical University, Guangzhou, China, 2 Key Laboratory of Neurogenetics and Channelopathies of Guangdong Province and the Ministry of Education of China, Guangzhou Medical University, Guangzhou, China, 3 Southern Medical University, Guangzhou, China, 4 Guangzhou Medical University, Guangzhou, China

¤a Current address: Mingxin Road #36, Liwan District, Guangzhou, China
¤b Current address: Shatai South Road#1023–1063, Baiyun District, Guangzhou, China
¤c Current address: Dongfeng West Road#195, Yuexiu District, Guangzhou, China
* zhouyllivy@aliyun.com (YZ); 32491329@qq.com (WL); yangwang2004@126.com (CY)

**Data Availability Statement:** No datasets were generated or analysed during the current study. All

## Abstract

### Background

Depression is a common psychological problem in adolescents worldwide. Although the World Health Organization recommends that members of this population engage in physical activity to reduce depressive symptoms, compliance with this recommendation is often low. Furthermore, although behavioral activation (BA) is recommended as a treatment for adolescents with depression, the reported effect size is small. Compared with traditional exercises, gamified physical activity (GPA) can be particularly appealing to adolescents because it is perceived as an enjoyable experience. In this study, we integrated BA and GPA to create behavioral activation play therapy (BAPT). We designed a clinical trial to investigate the feasibility, acceptability, and effectiveness of this treatment in adolescents with depression.

### Methods

This study is a randomized controlled trial (RCT) with a three-arm, assessor-blinded design, conducted to validate the effectiveness and applicability of BAPT for treating adolescent with depression. We will recruit 258 participants and randomly assign them to a BAPT group, BA group, or GPA group using a ratio of 1:1:1. Based on conventional strategies for treatment and care, the three groups will receive nine BAPT sessions, nine BA sessions, or nine GPA sessions, respectively. We will compare the outcomes of the BAPT with those of the BA and GPA interventions.

relevant data from this study will be made available upon study completion.

**Funding:** This study was supported by the Science and Technology Program of Guangzhou (grant No. 205171098044), Guangzhou Municipal Key Discipline in Medicine (2021-2023), Guangzhou High-level Clinical Key Specialty, and the Guangzhou Research-oriented Hospital. The funders had no role in study design, data collection and analysis, decision to publish, or preparation of the manuscript.

**Competing interests:** The authors have declared that no competing interests exist.

**Abbreviations:** MDD, Major depressive disorder; BA, behavioral activation therapy; GPA, gamified physical activities; BAPT, behavioral activation play therapy; RCT, randomized controlled trial; CBT, cognitive behavioral therapy; WHO, World Health Organization; PA, physical activity; DCAP, Department of Child and Adolescent Psychiatry; BTCAQ, Behavioral Therapist Competency/ Adherence Questionnaires; MADRS, Montgomery-Asberg Depression Rating Scale; ANSAQ, Adolescent Non-suicidal Self-injury Assessment Questionnaire; BDI-II-C, Beck Depression Inventory-II-Chinese version; BADS-SF, Behavioral Activation for Depression Scale Short Form; ISI, Insomnia Severity Index; IPAQ-SF, International Physical Activity Questionnaire-Short Forms; SPAP, Survey of Physical Activity Preference; SIAN, Short Inventory of the Assessment of Negative Effects; NEQ, Negative Effects Questionnaire; MET, metabolic equivalent; CRF, Case Report Form; SMD, standardized mean difference.

## Discussion

This is the first RCT to explore the effectiveness and applicability of BAPT in adolescents with depression. This study will provide evidence that may help to decrease depressive symptoms in adolescents, and will demonstrate the treatment effectiveness in terms of increasing levels of physical activity, reducing the rate of non-suicidal self-injury behaviors, and improving sleep quality. We will also assess the presence of side effects and the treatment adherence of patients receiving BAPT.

## Trial registration

**Trial registration:** Chinese Clinical Trial Registry, ChiCTR2300072671. Registered on 20 June 2023.

## Background

From 2012 to 2022, the incidence of depression in adolescents increased dramatically [1]. In 2020, 17.2% and 7.4% of adolescents in China were found to have mild and severe depressive symptoms, respectively, and the lifetime prevalence is expected to be 11%–20% [2]. The high prevalence of depression in adolescents is a public health issue because it can interrupt the developmental process and have negative corresponding effects throughout the lifespan [3, 4]. Adolescent major depressive disorder (MDD) can have serious consequences such as withdrawal from of school, drug use, self-harm, and suicide, and it is the main cause of illness and disability among adolescents. Currently, the World Health Organization (WHO) recommends psychotherapy as the first-line treatment for depression in adolescents, (note that antidepressants is not the recommended primary approach [5]. However, psychotherapy is associated with several drawbacks, as it can be difficult to access, costly, and minimally effective. According to some meta-analyses, the overall effect size (hedges' g) of psychotherapy for adolescent depression is small (g = 0.36, 95% CI: 0.25–0.47) [6], with a high dropout rate of 14.6% (95% CI 12.0, 17.4) [7]. Furthermore, some medications prescribed to treat adolescent MDD, such as fluoxetine and venlafaxine, have unfortunate side effects in adolescents including an increase in suicidal thoughts and behaviors. As a result of these drawbacks, some adolescents with depression do not benefit from medication or psychotherapy [8]. Accordingly, effective interventions for this population are an important focus for mental health researchers around the world [9, 10].

In recent years, exercise therapy has attracted global attention because of its simplicity and robust effects [11]. For instance, the effects exercise of a high or moderate intensity have been shown to be comparable to those of antidepressant treatment [12]. Consequently, clinicians have adopted an increasingly positive attitude towards the use of exercise in treating adolescents with MDD [13]. In 2020, the WHO recommended that children and adolescents engage in at least one hour of moderate-to-high physical activity (PA) every day to improve health and reduce symptoms of depression [14]. However, research has shown that only 20% of children and adolescents aged 13–15 comply with this recommendation [15]. Furthermore, adolescents with depression are less likely to engage in exercise, and depressive symptoms have been found to reduce the motivation to exercise [16]. Therefore, more effective treatment strategies are urgently needed.

In 2023, the WHO recommended behavioral activation (BA) as an effective form of psychotherapy for the treatment of depression [5]. BA is clinically as effective as cognitive behavioral therapy (CBT), costs up to 21% less [17], and is simple and easy to administer. Although BA is an effective treatment for adolescents with depression, it is difficult to administer in the adolescent population, leading to unsatisfactory outcomes [18]. As a result, health practitioners have attempted to apply BA in combination with other therapeutic techniques to obtain better treatment results [19–21]. For example, the combination of BA with digital networking technology was well received by adolescents [22–25]. However, that program led to an increase in screen time, Besides, which has been associated with decreased social and physical activity as well as increased sedentary behavior [26]. Furthermore, the risk of depression was found to increase in individuals who exceeded two hours of screen time per day [27].

BA including PA has been found to improve mood, which is beneficial for patients with depression, along with increasing the amount of regular exercise [28]. Studies have shown that BA interventions combined with PA have a high completion rate, and that they can significantly reduce depressive symptoms in patients [29]. BA and PA are highly consistent in terms of techniques, such as self-monitoring, goal setting, and problem-solving. Combining these two approaches could overcome the limitations associated with using each intervention alone, as well as reduce the recurrence rate of depression after treatment [30].

Although studies have demonstrated that BA combined with PA is feasible and acceptable for improving depressive symptoms in adults, relatively fewer studies have been conducted in adolescents. This may be because different age groups choose different forms of PA. In contrast to traditional forms of PA, adolescents are more inclined to select game forms of PA [31]. Therefore, the degree to which physical activities are perceived as pleasurable is an important factor to consider when developing therapies for use in this population [32]. Studies have shown that using gamification to promote physical activity can lead to better uptake [33]. Thus, interventions that focus on gamified physical activity (GPA) may be optimal for reducing sedentary behavior and depression risk in adolescents [34]. In most adolescents, play is a vital element of PA interventions, and the integration of play-centered activities is likely to be an effective method for increasing the level of PA and improving social connections. Indeed, encouraging adolescents to participate in active, stimulating, and adventurous play could enable them to test their abilities independently, thus improving social resilience [35–37]. Additionally, integrating competition and rewards may help to promote physical activity, along with other more innovative approaches [38].

## Study objectives

To date, no empirical studies have verified the effects of interventions that integrate GPA into BA with the goal of improving depressive symptoms in adolescents with depression in clinical settings in China. As a result, there are no guidelines regarding the exercise parameters necessary to produce anti-depressive effects. To address this, our research group integrated BA and GPA into behavioral activation play therapy (BAPT), and developed a randomized controlled trial (RCT) to evaluate the efficacy of BAPT for treating adolescents with depression. We plan to assess the effectiveness of BAPT in terms of increasing levels of physical activity, reducing nonsuicidal self-injury behavior, and ameliorating sleep quality. We also hope to discern the adaptability of this treatment with respect to clinical applications. We hypothesize that BAPT will be more effective than BA or GPA in relieving depressive symptoms and enhancing treatment adherence in adolescents with depression, with fewer side effects and better adherence rates.

## Methods and design

### Study design and setting

This study is a three-arm, assessor-blinded RCT. Participants will be randomly assigned to the BAPT group, the BA group, or the GPA group using a ratio of 1:1:1.

The study protocol is consistent with the Standard Protocol Items: Recommendations for Intervention Trials (SPIRIT) (see S5 File). **Fig 1** presents the enrolment, intervention, and evaluation schedule for SPIRIT. **Fig 2** presents a flow chart of the study. The study will be supervised by the Department of Child and Adolescent Psychiatry (DCAP) and the Academic Management Committee of the Affiliated Brain Hospital of Guangzhou Medical University.

### Recruitment

The recruitment process will continue until a total of 258 participants have been enrolled. Recruitment will be announced via flyers and posters in both inpatient and outpatient clinics, as well as online platforms associated with the Affiliated Brain Hospital of Guangzhou Medical University. Parents or legal guardians of potential participants will be invited to initiate contact with the research assistant via phone or WeChat to inquire about the study. Subsequently, a doctor will assess the eligibility of potential participants through video conferencing or face-to-face interviews. The doctor will employ the Structured Clinical Interview for DSM-5 Disorders-Clinician Version (SCID-5-CV). Measures will be implemented to ensure data confidentiality and security throughout the recruitment process.

### Participant eligibility

**Table 1** shows the inclusion and exclusion criteria for adolescents with depression.

**Therapist eligibility.**   The therapists will be recruited from the clinical staff of the Affiliated Brain Hospital of Guangzhou Medical University. Inclusion criteria: non-psychology healthcare professionals; bachelor's degree or higher; more than 1 year working in clinical psychiatry.

### Ethics statement

This study (see S1 and S2 Files) was approved by the ethics committee of the Affiliated Brain Hospital of Guangzhou Medical University on April 27, 2023 (Ethics number: 2023 [27], see S3 and S4 Files) and registered with the China Clinical Trials Center under the registration number ChiCTR2300072671.

**Assent and informed consent.**   Prior to enrollment in this study, the participants and their parents or legal guardians will be fully informed by the researchers about the purpose, procedures, risks, benefits and confidentiality of the study using plain language. Any inquiries will be addressed to ensure full comprehension of the study protocol. When participants provide their assent to participate in this study, their parents or legal guardians will sign an informed consent form. The participants will retain the right to withdraw from the study at any point without penalty, and their personal information will be handled in compliance with relevant privacy regulations. All data will be anonymized and securely stored. The interventions provided within the study will be offered free of charge. Upon completion of all interventions and assessments, the participants will receive an electronic exercise wristband as a reward.

**Self-injury and suicidal ideation..**   Before each intervention session, the participants will complete the 10th item of the clinician-rated Montgomery-Asberg Depression Rating Scale (MADRS) to assess suicidal risk. If a participant obtains a score >4, they will be temporarily

| | STUDY PERIOD | | | | | | | | |
| | Enrolment | Allocation | Post-allocation | | | | Close-out | | |
| TIMEPOINT** | -7 days | Week 0 | Week 1 | Week 2 | Week 3 | Week 4 | Week 8 | Week 16 | Week 28 |
| **ENROLMENT:** | | | | | | | | | |
| **Eligibility screen** | X | | | | | | | | |
| **Informed consent** | X | | | | | | | | |
| **Demographic** | X | | | | | | | | |
| **Randomization and allocation** | | X | | | | | | | |
| **INTERVENTIONS:** | | | | | | | | | |
| *[Behavioral Activation Play Therapy (BAPT)]* | | | ◄━━━━━━━━━━━━━► | | | | | | |
| *[Behavioral Activation (BA)]* | | | ◄━━━━━━━━━━━━━► | | | | | | |
| *[gamified physical activities (GPA)]* | | | ◄━━━━━━━━━━━━━► | | | | | | |
| **ASSESSMENTS:** | | | | | | | | | |
| *MADRS* | | X | | | | X | X | X | X |
| *BDI-II-C* | | X | | | | X | X | X | X |
| *BADS-SF* | | X | | | | X | X | X | X |
| *ANSAQ* | | X | | | | X | X | X | X |
| *ISI* | | X | | | | X | X | X | X |
| *IPAQ-SF* | | X | | | | X | X | X | X |
| *CACQ* | | | | | | X | | | |
| *SPAP* | | X | | | | | | | |
| *SIAN* | | | X | X | X | X | | | |
| *NEQ* | | | | | | | X | | |
| *Physical activity intensity* | | X | X | X | X | X | | | |

**Fig 1. Enrolment, intervention, and evaluation schedule for SPIRIT. Abbreviations:** *BMI*: Body Mass Index; *MADRS*: Montomery-Asberg Depression Rating Scale; *BDI-I-C*: Beck Depression Inventory-II-Chinese version; *BADS-SF*: Behavioral Activation for Depression Scale Short Form; *ANSAQ*: Adolescent Non-suicidal Self-injury Assessment Questionnaire; *ISI*: Insomnia Severity Index; *IPAQ-SF*: International Physical Activity Scale Short Form; *CACQ:* Custom Acceptance and Compliance Questionnaire; *SPAP*: Survey of Physical Activity Preference; *SIAN*: Short Inventory of the Assessment of Negative Effects; *NEQ*: Negative Effects Questionnaire.

withdrawn from the study. During the intervention, a research assistant will closely monitor mood changes in the participants. In the event of severe self-injury or suicidal behavior, participants will be temporarily withdrawn from the study. Following such occurrences, doctors

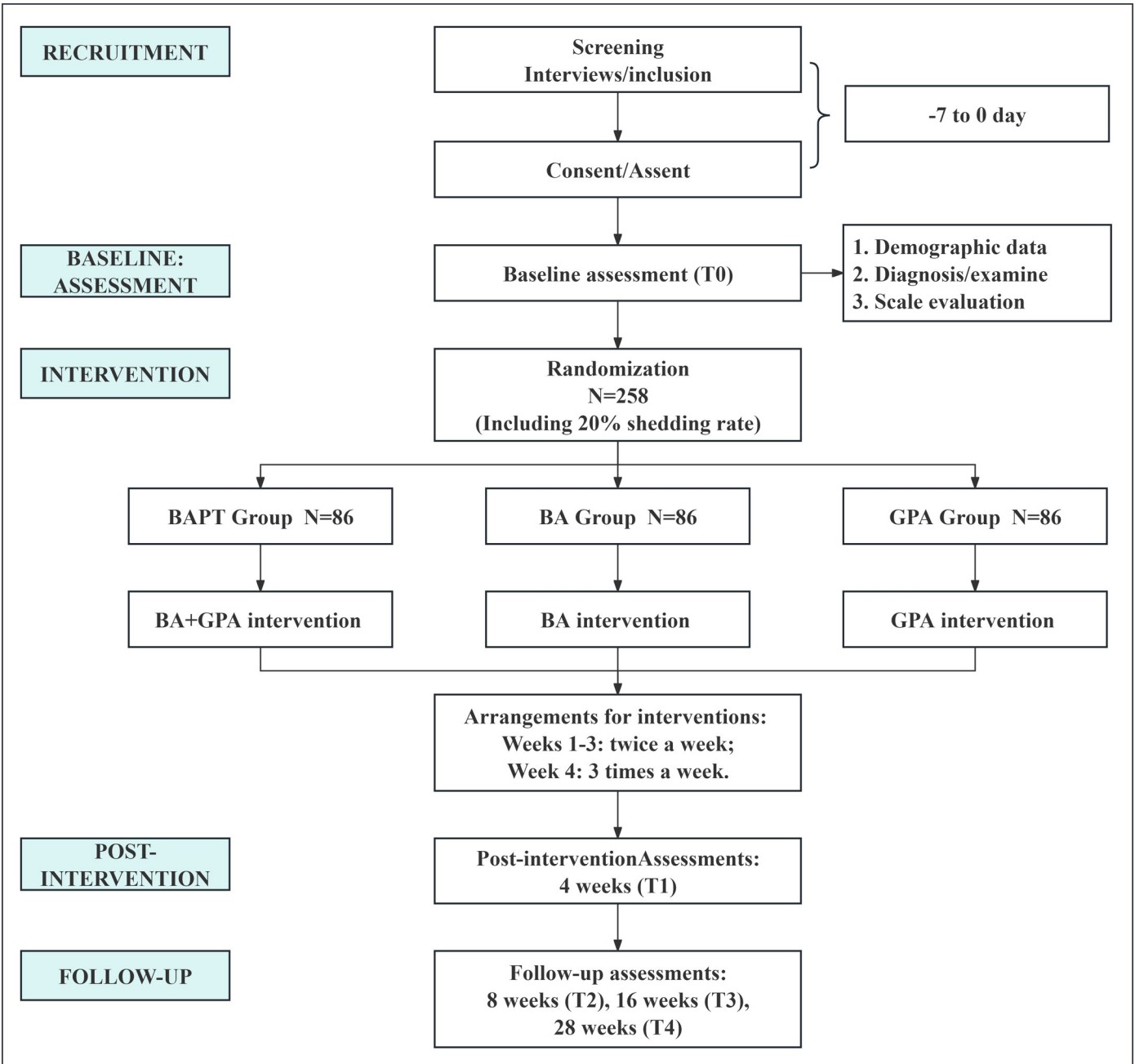

**Fig 2. Research flow chart.** *Participant data regarding physical intensity will be collected every week during the study.

within the study group will administer suicide crisis interventions or psychological counseling services. Additionally, expert discussions will be organized to prevent adverse incidents from occurring in the future.

**Risks associated with physical activity.** The study intervention involves physical activity. To ensure the safety of participants, we conducted preliminary testing for all GPA protocols. Although our test results indicate that the GPA is rated as moderate to high intensity physical activity, it remains safe. We mapped out a complete risk prevention scenario that was

**Table 1. Inclusion and exclusion criteria.**

| Subject eligibility | |
|---|---|
| **Inclusion criteria** | 1. DSM-5-based diagnosis of depression without psychotic features;<br>2. MADRS score $\geq$ 12;<br>3. Inpatient aged 12–17 years;<br>4. Intelligence level in the normal range, normal perception and self-expression ability, and ability to complete the baseline scale assessment;<br>5. Informed consent form signed by the participant and their legal guardians. |
| **Exclusion criteria** | 1. DSM-5 diagnosis of other mental disorders, including addictive disorders, developmental disorders, bipolar disorder, substance-related disorders, and schizophrenia (except anxiety disorders);<br>2. Severely disruptive or aggressive behaviors, or positive suicidal ideation (suicide item score on the MADRS > 4);<br>3. Severe psychotic symptoms (presence of pain or common hallucinations and/or delusions);<br>4. Clinically significant and uncontrolled pulmonary, endocrine, immunological, or cardiovascular diseases (based on ancillary examination, physical examination, and medical history);<br>5. Inability to cooperate with the cognitive function tests or ineligible for the study for other reasons.<br>6. Currently undergoing other forms of psychotherapy. |

**Abbreviations**: *DSM-5*: Diagnostic and Statistical Manual of Mental Disorders; *MADRS*: Montgomery-Asberg Depression Rating Scale.

approved by the ethics committee of the Affiliated Brain Hospital of Guangzhou Medical University. If an adverse event occurs, the study group will manage it appropriately, document it comprehensively, and report it to the Ethics Committee and the Academic Management Committee.

## Intervention fidelity

To enhance the fidelity and replicability of the interventions, we have developed separate manuals for the BAPT, BA, and GPA interventions. These are intended for therapist reference and training program. All therapists will undergo a two-week training program, and the effectiveness of the training will be evaluated using the Behavioral Therapist Competency/Adherence Questionnaire (BTCAQ) which was adapted from Weisman et al. [39]. The BTCAQ includes 13 5-point Likert items distributed among six major categories: 1. Education (comprehension of the principles and objectives of the course); 2. Group leadership skills (capacity to issue instructions, guide implementation, communicate effectively, and offer feedback); 3. Assessment skills (ability to facilitate brainstorming, evaluate ideas, and assess problem-solving); 4. General skills (proficiency in building rapport, efficiently managing time, and assigning homework); 5. Provision of a secure and efficacious therapeutic environment; 6. Evaluation of participants' adherence to the intervention. The therapists will be required to obtain a certificate issued by the Affiliated Brain Hospital of Guangzhou Medical University via an assessment before implementing the interventions. Additionally, all therapists will receive ongoing support through weekly online clinical supervision sessions conducted by expert mentors responsible for teaching these courses. With consent from both the therapists and participants, the researchers will randomly select two sessions during each intervention period and utilize the BTCAQ to evaluate the fidelity, adherence, feasibility, and acceptability of the intervention plan.

## Randomization

Random number will be generated by a specially recruited staff member using the SPSS Rv. Uniform function, and added to a random number table. The random seed parameters and

groupings will be sealed as confidential data in an opaque envelope. The random sequence and grouping data will be kept by a designated person, who will open the envelope after individuals meeting the criteria are selected to enter the study and have signed the informed consent form. The investigator will be informed regarding the participants' treatment allocation, and the participants will be randomized according to a pre-defined randomization protocol.

## Blinding

This study will adopt an assessor-blinded approach, as blinding the participants and interventionists to the study conditions is not feasible within this behavioral intervention. To minimize the impact of information bias, five evaluators responsible for measuring study outcomes will be recruited from external sources and kept unaware of the group assignments. To ensure the blinding of the evaluators to the group assignments, they will not conduct follow-up measurements for the same group. Fig 3 presents the evaluation arrangements. Prior to study commencement, all evaluators will receive consistency training. If any contingencies arise during the intervention, the blinding will be unveiled.

## Intervention

The BAPT, BA, and GPA groups will each receive 9 sessions conducted in a group format over a 4-week period. Each session will last for 60 minutes, with 2 sessions per week for the first 3 weeks and the last 3 sessions completed in week 4. During the study period, the participants will only receive conventional clinical medication and care, and will not be involved in any other psychological therapy programs.

All participants will use electronic exercise wristbands (Beijing Xiaomi Technology Co., LTD., Product number: Xiaomi Band 7pro, M2140B1) to record the daily intensity of their physical activities. The electronic exercise wristbands utilize photocapacitance pulse wave tracing (PPG) and acceleration sensors to measure the intensity of physical activity in adolescents [40–43].

**BAPT group.** Each session will consist of BA (30 minutes) and GPA (30 minutes). The BA procedure was adapted from the protocols of Kellett et al. [44] and Lejuez et al. [45], with appropriate modifications according to the present study setting. In the BA sessions,

| TIMEPOINT** | EVALUATION ARRANGEMENTS | | | | |
|---|---|---|---|---|---|
| | Week 0 | Week 4 | Week 8 | Week 16 | Week 28 |
| BAPT Group | A | B | C | D | E |
| BA Group | B | C | D | E | A |
| GPA Group | C | D | E | A | B |

*A, B, C, D, E represent 5 independent evaluators.

**Fig 3. Evaluation arrangements.** *A, B, C, D, E represent 5 independent evaluators.

participants will receive information about the core principles of BA, which can be summarized as nine topics as follows: (1) Initiation of BA: the program values and sense of connection; (2) the BA model and emotional monitoring; (3) get motivated: goal-oriented behavior; (4) get activated: situation-action-emotion; (5) problem solving skills; (6) goal setting and adjustment; (7) identifying obstacles and overcoming avoidance; (8) thinking: worrying, ruminating; (9) prevention of recurrence. Each session will contain one topic, and the GPA associated with that topic will be completed after that part of the BA. The BAPT scheme is shown in Table 2. All GPAs were designed or modified by the research team to be competitive, adventurous, interactive, and to follow certain rules that are in line with the physical and mental development characteristics of adolescents [31]. For example, "Cat catches mouse" involves the identification of obstacles and overcoming avoidance (corresponding to topic 7). It is an exciting chase game in which participants are asked to choose one of three roles: one pair of participants plays as the cat and mouse, while the others are fences. All the fences form a circle, and standing about 1 meter apart. The mouse is inside the circle and the cat is outside, and when the game starts, the cat attempts to enter the circle to catch the mouse. To avoid being captured, the mouse can choose a fence by touching that participant. Then, the original fence becomes the cat, the original cat becomes a mouse, and the cycle continues until the cat catches the mouse. The game is interesting and challenging because of the constantly changing roles of the participants in the game. The researchers rated all GPAs as involving moderate to high intensity physical activity after testing with electronic exercise wristbands.

After each session, the participants will receive homework. They will be encouraged to schedule activities (socio-motor games, etc.) that feel enjoyable, rewarding, and offer a sense of control during the following week, and to monitor the impact of these activities on their personal mood. Homework will be collected by the researchers once a week.

**BA group.** The BA group intervention will be same as the BA in the BAPT group, but without the GPA. Each session will be 60 minutes long because of the extended discussion. The homework will be the same as in the BAPT group, but without the socio-motor game suggestions.

**GPA group..** The GPA group intervention will be same as the GPA in the BAPT group, but without the BA. Each session will last 60 minutes and will include an explanation of the GPA regulations, GPA intervention, and guidance on monitoring mood changes before and after the intervention. The homework will be the same as in the BAPT group.

### Efficacy evaluations

From week 0 to week 28, the researchers will utilize MADRS scores as a primary outcome for assessing changes in depressive symptoms. A total score of 0–60 is obtained by completing 10

**Table 2. The scheme of the BAPT.**

| Time | Session | Course theory | Game-type physical activity |
|---|---|---|---|
| Week 1 | 1 | Initiate BA: values and the sense of connection | Strong Winds Blow |
| | 2 | The BA model and emotional monitoring | Balloon Missile |
| Week 2 | 3 | Get motivated: goal-oriented behavior | Giant's Hat |
| | 4 | Get activated: situation-action-emotion | Overcome The Obstacle |
| Week 3 | 5 | Problem solving skills | Transmitting Things by Sole |
| | 6 | Goal setting and adjustment | Blindfolded Communication |
| Week 4 | 7 | Identify obstacles and overcome avoidance | Cat Catches Mouse |
| | 8 | Thinking: worrying, ruminating | Knee Pat |
| | 9 | Prevention of recurrence | Queen Ant Game |

items with scores ranging from 0 to 6 [46]. The depressive symptoms will be grouped as extreme, major, moderate, or mild according to the following scores: MADRS > 35, $30 \leq MADRS < 35$, $22 \leq MADRS < 30$, and $12 \leq MADRS < 22$, respectively, and MADRS < 12 indicates no depressive symptoms. Compared with the baseline, a MADRS total score reduction of 50% will be defined as a significant antidepressant response, while a reduction of 20% will be defined as an improvement.

Secondary outcome measures will include: (1) Self-rated depressive symptoms on the Beck Depression Inventory-II-Chinese version (BDI-II-C): This scale has high internal consistency, and reflects the severity of depression over the past two weeks. It contains 21 items, each rated on a 0–3 scale, with an overall score of 0–63 points. The levels of depression will be grouped as severe, moderate, or mild according to the following score ranges: 29–63, 20–28, and 14–19, respectively, and a score of 0–13 will be classified as no depression [47]. (2) Behavioral Activation for Depression Scale Short Form (BADS-SF): This scale consists of nine items that measure changes in behavioral activation during the past week, including the measurement day. The activation (AC) subscale includes questions 1, 2, 3, 4, 5, and 9, while the avoidance (AV) subscale includes questions 6, 7, and 8. The scale uses a 7-level scoring method, which ranges from 0 (not at all) to 6 (completely). A higher score for an item represents an answer that is closer to the item statement. The scale has been demonstrated to have robust reliability and validity [48] (3) Adolescent Non-suicidal Self-injury Assessment Questionnaire (ANSAQ): This questionnaire is divided into two parts: a behavioral questionnaire (12 items) and a functional questionnaire (19 items) to evaluate self-injury behavior [49]. It uses the Likert 5-point scale, where "1 to 5" corresponds with "no, occasionally, sometimes, often, always", respectively. A higher score represents a more serious degree of self-injury. The questionnaire has high internal consistency, while that of the behavioral questionnaire was 0.921. In this study, we will only use the behavioral dimension of the questionnaire (12 questions in total) to assess whether the adolescents engage in self-injury behavior. (4) Insomnia Severity Index (ISI): this is used to evaluate the severity of insomnia in patients. Total scores of 0–28 are obtained according to responses to 7 items with scores ranging from 0 to 4 [50]. According to the scoring guide, scores for mild, moderate, and severe clinical insomnia range from 8–14, 15–21, and 22–28, respectively, while scores less than 7 are considered to represent non-clinical insomnia. (5) International Physical Activity Questionnaire-Short Forms (IPAQ-SF): This short form records activity for four intensity levels: 1) vigorous-intensity activity such as aerobics, 2) moderate-intensity activity such as leisure cycling, 3) walking, and 4) sitting [51]. These four intensity levels (except sitting) can be determined and reported as MET-minutes per week. The IPAQ-SF has good internal consistency. (6) Custom Acceptance and Compliance Questionnaire (CACQ): This questionnaire is designed to assess acceptance and compliance and comprises 7 items rated on a 5-point Likert scale. It evaluates participants' satisfaction, intention to continue, and intention to recommend the service. Additionally, adherence rates will be analyzed. (7) Survey of Physical Activity Preference (SPAP): This survey will be used to investigate the physical activity preferences of adolescents with depression at baseline. The outline of the interview is as follows: 1) How do you perceive the importance of physical activity in your daily life? 2) What different types of physical activities have you tried before? 3) What types of physical activities do you enjoy? 4) What is your favorite physical activity and why? 5) What goals do you hope to achieve through physical activity? (8) Short Inventory of the Assessment of Negative Effects (SIAN) [52]: This scale assesses the negative effects of each intervention, and contains 7 items on a 4-point Likert scale ("strongly agree" to "strongly disagree"). The items include frustration with treatment, increased depression, crisis, rumination, no longer liking oneself, feeling of being burdened by treatment, fear of attending treatment. The internal consistency of the SIAN was good, with a Cronbach's alpha = 0.823.

(9) Negative Effects Questionnaire (NEQ) [53]: This is a self-report instrument containing 20 items representing adverse or unwanted events. First, the patients rate the occurrence of specific events, and report whether they occurred during treatment (yes/no). Second, they indicate how negatively the events affected them on a 5-point Likert scale (0–4). Last, the participants will be asked to identify the negative effect.

At the baseline, week 4, week 8, week 16 and week 28, the participants from the three groups will be evaluated with the corresponding scales to assess the degree of depression, behavioral activation level, non-suicidal self-injury behavior, and sleep quality scores. The SIAN and NEQ will be evaluated weekly and at week 16 respectively. The CACQ will be evaluated at week 4. We will also calculate the improvement of symptoms for each dimension in each group at each time point, and then compare the divergences between the three groups.

**Demographic and clinical data..** Demographic data (age, sex, occupation, culture, residence, family structure, economic income, etc.), treatment history, developmental history, history of tobacco, alcohol and other psychoactive substance use, previous receipt of psychological intervention (including the type, frequency, and duration), psychiatric symptoms, and length of hospital stay will be collected during the screening stage.

**Auxiliary inspection..** Electrocardiograms, routine blood analysis, and biochemical indexes of clinical treatment will be completed at the baseline assessment.

**Physical activity intensity..** The Metabolic Equivalent (MET) can be used to express the relative energy metabolism corresponding to specific levels of physical activity. Depending on whether intensity of physical activity is light, medium, or heavy, it will be represented by a score of 0–3, 3–6, and more than 6, respectively.

The researchers will collect data regarding the participants' physical activity intensity once a week. Participants who complete the homework and exercise at a moderate intensity for 150 minutes or above per week will be rewarded with small gifts for positive reinforcement. The electronic exercise wristbands will be purchased and distributed by the project team to ensure monitoring consistency.

## Data management

The Case Report Form (CRF) will be utilized to document the demographic information and clinical symptoms of all participants, which will subsequently be stored on the Clinical Trial Management Public Platform (URL: http://www.medresman.org.cn/) through a process of double data entry. The primary responsibility of the study leader will be to ensure the integrity, accuracy, and promptness of the data entry process. To maintain the privacy of the participants, all files will be protected by password and will only be accessed by the relevant members of research group. The researchers will remove the names, phone numbers, and addresses that are not related to the study during the data entry process. Numbers will be used to identify the participants. In addition, the original CRF will be securely stored in a locked filing cabinet throughout the study, accessible only to the project leader. A dedicated physician (Prof. Yanling Zhou) will monitor the wholeness, uniformity, and plausibility of the data.

## Sample size calculation

The sample size in this study will be calculated using a one-way analysis of variance (ANOVA) with an effect size of f = 0.25, a bilateral $\alpha$ = 0.05, and a 90% confidence interval. We used G*Power software to ascertain that a total sample size of 207 is required. Assuming a dropout rate of 20%, approximately 258 participants will be needed in total, with 86 participants in each group.

## Data analysis

This study is based on the intention-to-treat principle. SPSS 28.0 will be used for data analysis. First, to ensure comparability after randomization, we will compare the differences between the three groups at the baseline. The continuous variables will be analyzed by a t-test when the data have a normal distribution, and non-normally distributed data will be analyzed using the Kruskal-Wallis H test. Baseline differences between the three groups in terms of sociodemographic and clinical variables will be assessed using Chi-square tests for categorical variables and an ANOVA for continuous data. A one-way ANOVA will be used to identify variables (e.g., socio-demographic information, etc.) that have statistically significant differences ($p < 0.05$) among the three groups at baseline. These potential confounders will then be included as covariates in the ANOVA model for analysis. Post-hoc comparisons will be conducted using the Bonferroni method. To compare the intervention effects among the groups, a mixed-effect regression model will be used. Group (BAPT, BA, or GPA) will be entered as an inter-subject factor, and time (baseline, post-treatment, or follow-up) will be entered as an intra-subject factor. We will utilize Cohen's d for both the intra- and intergroup analyses.

## Discussion

A close relationship between physical activities and depression has been repeatedly confirmed by numerous studies. Kandola et al. used objective measures to find that declining levels of light activity and increasing periods of sedentary behavior between the ages of 12 and 16 years were related to greater depressive symptoms at age 18 years. Compared with those with consistently low levels of sedentary behavior, depression scores were 24.9%–28.2% higher in those with consistently high levels of sedentary behavior. Increasing light activity by one hour every day has previously been found to decrease depression scores by 8–11%, and sustained heavy physical activity was also related to lower depression scores [54]. Furthermore, PA, whether low, moderate, or high intensity, was found to be effective in treating mild to moderate depression [55]. Although physical activity can boost mental health, a pooled analysis of 16 million adolescents in more than 140 countries showed low rates of participation in physical activity among adolescents [56], such that nearly 85% of young people in China lacked adequate physical activity. Only approximately 20% of students tested met the standard of one hour of PA per day. In addition, PA levels in adolescents tend to decline with age because of excessive academic pressure, a lack of interest in sports and self-exercise habits, inadequate guidance, changes in leisure and entertainment methods, and insufficient sports venues and equipment [57]. Particularly, adolescents with depression reflected by persistent low mood, loss of interest, and other disease symptoms had lower adherence to physical activity guidelines [16].

Many guidelines recommend psychotherapy as an evidence-based treatment for adolescents with depression [58, 59]. Psychotherapy in this group has been found to have a significant effect compared with a control group, but the effect size was relatively small [6, 60, 61]. Specifically, a meta-analysis of the effectiveness of psychotherapy for adolescents with depression showed that the overall response rates to psychotherapy at 1 and 3 months after baseline were only 39% (95% CI: 34–45) and 24% (95% CI: 0.19–28), respectively. Notably, over 60% of adolescents who had received relevant psychological treatment did not respond to the treatment [18]. Another meta-analysis of the effects of psychological treatment in individuals of all ages with depression noted that the effect size was relatively small in adolescents, especially those younger than 13 years, compared with adults [62]. A recent systematic review and meta-analysis of the long-term effects of psychosocial interventions for adolescents with depression, involving 46,678 participants, found that interventions targeting adolescents specifically resulted in small but positive effects. The standardized mean difference (SMD) was -0.08 (95%

CI: -0.20 to -0.03, p = 0.002, $I^2$ = 72%) and -0.12 (95% CI: -0.20 to -0.03, p = 0.01, I2 = 68%) at 1 and 2 years, respectively [63]. Despite the best efforts of researchers, the current interventions do not adequately satisfy the needs of adolescents with depression [64]. This may be the reason for the small therapeutic effect. Adolescents habitually use behavioral strategies (rather than cognitive ones) to alleviate depression [65]. A meta-analysis using data from 4 RCTs showed that the pooled SMD for a BA intervention was 0.7, (95% CI -1.20, -0.20), with high heterogeneity ($I^2$ = 0.79) [66]. These findings indicate that to further reduce the burden of adolescents with depression, effective and innovative treatments are essential.

Research has shown that combining BA with improvisation in the form of games can improve adolescents' motivation regarding therapy, thereby decreasing depressive symptoms [19]. Compared with improvisation-based activity, our GPA may be more appropriate for adolescents with depression. This is because the GPA incorporates elements of gaming, games with rules, risk-taking, stimulation, and competition, which are consistent with adolescent psychological development [31], and also includes moderate to high intensity physical activities, which align with the WHO's recommendations for physical activity in adolescents [14]. Our preliminary exploration indicated potential acceptability of the GPA among adolescents with depression, which led us to initiate this study. BAPT is an innovative intervention that includes both psychological intervention techniques and GPA, making it appropriate for adolescents with depression. We anticipate that this study will determine the feasibility and acceptability of BAPT as a treatment for adolescents with depression. In this study, we will compare the side effects and adherence rates of BAPT, BA, and GPA in adolescents with depression. Furthermore, we will compare the effectiveness of BAPT, BA, and GPA in improving depressive symptoms, altering behavioral activation levels, and influencing sleep quality, as well as decreasing non-suicidal self-injury in adolescents. This will provide important information about the potential clinical value of BAPT.

## Advantages and limitations

This is a preliminary investigation of the effects of a new innovative intervention for depression. However, there are some limitations to this study. First, the study only compares BAPT with BA and GPA, and not with other standard treatments such as CBT, IPT, or medication. This will limit the degree to which we can speculate regarding the relative efficacy of BAPT. Second, this study involves behavioural interventions, that limit the use of a double-blind design. Therefore, only the assessors can be blinded,and there is potential for bias. Third, the participants will be solely recruited from inpatient and outpatient departments, which may limit the representativeness of the sample with respect to the broader population of adolescents with depression.

## Conclusion

This study examines the utility of BAPT in improving depression symptoms as an effective adjuvant therapy. We hope to provide evidence for the clinical implementation of a new mental health treatment for adolescents with depression. Future research will consider expanding recruitment to encompass more diverse settings such as communities and schools. Additionally, BAPT will be compared with other standard treatments to gain a better understanding of its true efficacy and applicability.

## Supporting information

**S1 File. Protocol for ethics committees.** The original protocol approved by The Regional Committee for Medical and Health Research Ethics.
(DOCX)

**S2 File. Protocol for ethics committees translate.** Translated version of the original protocol approved by the Regional Committee for Medical and Health Research Ethics.
(DOCX)

**S3 File. Protocol- ethics approval.** The letter of notification of the review opinion of the Regional Committee for Medical and Health Research Ethics.
(PDF)

**S4 File. Protocol-translate of ethics approval.** Translated version of the letter of notification of the review opinion of the Regional Committee for Medical and Health Research Ethics.
(PDF)

**S5 File. SPIRIT checklist.** Recommended items to address in a clinical trial protocol and related documents.
(DOCX)

## Acknowledgments

We would like to acknowledge Prof. Liang Zhou and Dr. Shao-Ling Zhong for their guidance in the early stages of the study, and TopEdit (www.topeditsci.com) for its linguistic assistance during the preparation of this manuscript.

## Author Contributions

**Conceptualization:** Xiaolong Huang.

**Data curation:** Yanling Zhou.

**Funding acquisition:** Chanjuan Yang, Wei Luo, Yanling Zhou.

**Investigation:** Xiaolong Huang, Yuqi Chen, Jiacheng Luo.

**Methodology:** Dongdong Wang.

**Project administration:** Xiaolong Huang, Yanling Zhou.

**Software:** Xiaolong Huang.

**Supervision:** Chanjuan Yang, Wei Luo, Yanling Zhou.

**Writing – original draft:** Xiaolong Huang.

**Writing – review & editing:** Yuqi Chen, Jiacheng Luo, Chanjuan Yang, Yanling Zhou.

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
