## [Decision Letter · Decision Letter 0]

27 Feb 2024

PONE-D-23-39367The effect of behavioral activation play therapy in adolescents with depression: a study protocol for a randomized controlled trialPLOS ONE

Dear Dr. Zhou,

Thank you for submitting your manuscript to PLOS ONE. After careful consideration, we feel that it has merit but does not fully meet PLOS ONE’s publication criteria as it currently stands. Therefore, we invite you to submit a revised version of the manuscript that addresses the points raised during the review process.

We look forward to receiving your revised manuscript.

Kind regards,

Dickens Akena, Ph.D

Academic Editor

PLOS ONE

Journal Requirements:

"This study was supported by Science and Technology Program of Guangzhou (grant No. 205171098044), Guangzhou Municipal Key Discipline in Medicine (2021-2023), Guangzhou High-level Clinical Key Specialty, and Guangzhou Research-oriented Hospital."

5. We note that the original protocol file you uploaded contains a confidentiality notice indicating that the protocol may not be shared publicly or be published. Please note, however, that the PLOS Editorial Policy requires that the original protocol be published alongside your manuscript in the event of acceptance. Please note that should your paper be accepted, all content including the protocol will be published under the Creative Commons Attribution (CC BY) 4.0 license, which means that it will be freely available online, and any third party is permitted to access, download, copy, distribute, and use these materials in any way, even commercially, with proper attribution.

Therefore, we ask that you please seek permission from the study sponsor or body imposing the restriction on sharing this document to publish this protocol under CC BY 4.0 if your work is accepted. We kindly ask that you upload a formal statement signed by an institutional representative clarifying whether you will be able to comply with this policy. Additionally, please upload a clean copy of the protocol with the confidentiality notice (and any copyrighted institutional logos or signatures) removed.

Additional Editor Comments:

The reviewers have provided their comments. You will need to address their comments and re-submit.

Reviewers' comments:

Reviewer's Responses to Questions

**Comments to the Author**

1. Does the manuscript provide a valid rationale for the proposed study, with clearly identified and justified research questions?

Reviewer #1: Yes

Reviewer #2: Yes

2. Is the protocol technically sound and planned in a manner that will lead to a meaningful outcome and allow testing the stated hypotheses?

Reviewer #1: Partly

Reviewer #2: Partly

3. Is the methodology feasible and described in sufficient detail to allow the work to be replicable?

Reviewer #1: No

Reviewer #2: Yes

4. Have the authors described where all data underlying the findings will be made available when the study is complete?

Reviewer #1: No

Reviewer #2: Yes

5. Is the manuscript presented in an intelligible fashion and written in standard English?

Reviewer #1: Yes

Reviewer #2: Yes

6. Review Comments to the Author

You may also provide optional suggestions and comments to authors that they might find helpful in planning their study.

Reviewer #1: Area of review Comment

• What are the main claims of the paper and how significant are they for the discipline?

1. Randomized Controlled Trial (RCT) Design: The RCT is a strong study design for determining the efficacy of interventions. However, the success of an RCT largely depends on its execution, including randomization methods, blinding, and control of confounding factors.

2. Single-blind Approach: The study used a single-blind approach, in which outcome assessors were unaware of the group assignment. Although this helps reduce bias, it is not as robust as a double-blind design. In interventions involving physical activity, double-blinding is challenging, but a single-blind design may still introduce bias.

3. Sample Size and Population: A sample size of 200 adolescents appears adequate, but it is important to consider whether this sample is representative of the broader adolescent population, which is not clearly stated in the article.

4. Intervention Complexity: The integration of BA and GPA could add complexity to the intervention, which might impact its replicability in different settings or practicality in routine clinical practice.

Potential Limitations and Areas for Improvement

1. Follow-Up Duration: The long-term effects of BAPT are not clear from the abstract. A longer follow-up period would be beneficial for assessing the sustainability of treatment effects.

2. Comparison with Existing Treatments: The study compared BAPT with BA only, and not with other standard treatments such as CBT or medication. These comparisons provide a clearer picture of BAPT's relative efficacy of BAPT.

3. Measurement of Outcomes: The Reliance on self-reported measures for some outcomes could introduce bias. Objective measures, where feasible, would strengthen this study's findings.

4. Consideration of Confounding Variables: Adolescent depression can be influenced by numerous social, environmental, and genetic factors. This study accounted for these potential confounders in the analysis.

Implications and Significance

1. Innovative Approach: This study's focus on integrating gamified physical activities into behavioural activation therapy is innovative and could lead to more engaging treatment modalities for adolescents.

2. Potential Clinical Impact: If BAPT is effective, it could offer a valuable tool for treating adolescent depression, potentially improving adherence and outcomes.

3. Contribution to the Field: This research could contribute significantly to the understanding of non-pharmacological interventions in adolescent mental health, an area that requires more empirical data.

While this study presents an innovative approach that has the potential to make significant contributions to the field of adolescent mental health, it is essential to carefully consider its design, execution, and potential limitations. These results should be interpreted within the context of these factors to understand the true efficacy and applicability of BAPT.

• Are the claims properly placed in the context of the previous literature? Have the authors treated the literature fairly?

Comparative Analysis: This study could be strengthened by explicitly comparing the proposed BAPT method with existing treatments in terms of efficacy, side effects, and adherence rates.

Broader Literature Scope: Depending on the full content of the paper, it would be beneficial to include a wider range of studies, particularly those with opposing viewpoints or results, to provide a more comprehensive view of the current state of research.

Longitudinal Studies and Meta-Analyses: The inclusion and discussion of longitudinal studies or meta-analyses in the field would provide deeper insights into the long-term effectiveness of treatments for adolescent depression

• Do the data and analyses fully support the claims? If not, what other evidence is required?

No data sets have been described

• PLOS ONE encourages authors to publish detailed protocols and algorithms as supporting information online. Do any particular methods used in the manuscript warrant such treatment? If a protocol is already provided, for example for a randomized controlled trial, are there any important deviations from it? If so, have the authors explained adequately why the deviations occurred? The authors must provide a detailed guide on how recruitment will be performed and explain how the fidelity, feasibility, and acceptability of the intervention will be achieved.

• If the paper is considered unsuitable for publication in its present form, does the study itself show sufficient potential that the authors should be encouraged to resubmit a revised version?

Upon addressing any outstanding queries and incorporating the responses, this paper will be deemed suitable for publication.

• Are original data deposited in appropriate repositories and accession/version numbers provided for genes, proteins, mutants, diseases, etc.?

None seen

• Does the study conform to any relevant guidelines such as CONSORT, MIAME, QUORUM, STROBE, and the Fort Lauderdale agreement?

Yes, it conforms to CONSORT

• Are details of the methodology sufficient to allow the experiments to be reproduced?

A more in-depth description of the methods is required.

• Is any software created by the authors freely available?

Not mentioned

• Is the manuscript well organized and written clearly enough to be accessible to non-specialists?

With minor edits to the language used, it will be suitable.

• Is it your opinion that this manuscript contains an NIH-defined experiment of Dual Use concern?

It is highly unlikely that this manuscript contains an NIH-defined experiment on dual-use research of concern.

Reviewer #2: "....On the basis of conventional treatment and care, we will 72 compare the outcomes of BAPT and BA. The intervention group and control group will 73 receive nine sessions of BAPT or BA, respectively"

Comment: Since the effect size of BA is already known to be very small, why not have three treatment arms comparing BAPT, BA, and GPA.

You need a dedicated section on ethical considerations in this protocol. Describe in detail how the double vulnerabilities of adolescence and depression will be handled to prevent further harm and how voluntariness will be protected. Also minors cannot give informed consent but rather assent, the protocol only talks about informed consent.

WHO defines adolescents as individuals between ages 10-19. Explain your choice of ages 12-17.

Specialised psychologists will be used to deliver the intervention. How do you eliminate the chances that there is psychological knowledge/skills transfer during the activities, such that at the end of the activities you can be sure that it is indeed BAPT and not interaction with the psychologists (in a novel, play setting) that is causing any effect recorded.

line 261-262: How will you control for the effect of possible psychological intervention received up until the point of hospitalisation and recruitment into the study?

Line 481-482: I think this is an over generalization of adolescents' wishes and preferences. It is assumed here that the adolescents are a homogenous group. Consider having some qualitative interviews with at least some of the participants concerning their views on physical activity.

7. PLOS authors have the option to publish the peer review history of their article (what does this mean?). If published, this will include your full peer review and any attached files.

Reviewer #1: No

Reviewer #2: **Yes: **LYNDA NAKALAWA

---

## [Author Response · Author response to Decision Letter 0]

12 Apr 2024

We appreciate the insightful comments provided by the editor and reviewers, and we acknowledge the constructive feedback they have offered. We have submitted three documents in response to the reviewers' comments: a revised manuscript incorporating the suggested changes, a tracked changes document illustrating the revisions made in relation to the original manuscript, and a comprehensive response letter addressing each comment raised by the editor and reviewers. In response to the editor and reviewers' insightful feedback, we have made significant revisions to the manuscript to address their concerns and improve clarity, coherence, and accuracy. The response letter comprehensively addresses each comment raised by the editor and reviewers, providing detailed explanations and clarifications where necessary. We are grateful for the editor and reviewers' dedication and guidance throughout the review process, and we look forward to continued collaboration and refinement of the manuscript.

---

## [Editor Report · Decision Letter 1]

7 May 2024

The effect of behavioral activation play therapy in adolescents with depression: a study protocol for a randomized controlled trial

PONE-D-23-39367R1

Dear Dr. Zhou

We’re pleased to inform you that your manuscript has been judged scientifically suitable for publication and will be formally accepted for publication once it meets all outstanding technical requirements.

Kind regards,

Dickens Akena, Ph.D

Academic Editor

PLOS ONE
---

## [Editor Report · Acceptance letter]

13 May 2024

PONE-D-23-39367R1 

PLOS ONE

Dear Dr. Zhou, 

I'm pleased to inform you that your manuscript has been deemed suitable for publication in PLOS ONE. Congratulations! Your manuscript is now being handed over to our production team.

Kind regards, 

on behalf of

Dr. Dickens Akena 

Academic Editor

PLOS ONE